# Are We the Problem? A Call to Action for Addressing Institutional Challenges to Engaging Community Partners in Research

**DOI:** 10.3390/ijerph21020236

**Published:** 2024-02-17

**Authors:** Neha Hippalgaonkar, Ryan Huu-Tuan Nguyen, Eliza Brumer Cohn, Joseph Horowitz, Ana Williams Waite, Tigist Mersha, Christen Sandoval, Sarah Khan, Kauthar Salum, Paris Thomas, Anne Marie Murphy, Beulah Brent, Lolita Coleman, Paramjeet Khosla, Kent F. Hoskins, Vida Henderson, Leslie R. Carnahan

**Affiliations:** 1Division of Hematology/Oncology, University of Illinois Chicago, Chicago, IL 60612, USA; rnguye8@uic.edu (R.H.-T.N.); khoski@uic.edu (K.F.H.); 2University of Illinois Cancer Center, Chicago, IL 60612, USA; awilli97@uic.edu (A.W.W.); tmersh2@uic.edu (T.M.); lcarna2@uic.edu (L.R.C.); 3Fred Hutchinson Cancer Center, Seattle, WA 98126, USA; ecohn@fredhutch.org (E.B.C.); ksalum@fredhutch.org (K.S.); vahender@fredhutch.org (V.H.); 4Department of Medicine and Pediatrics, University of Illinois College of Medicine, Chicago, IL 60612, USA; jhorow4@uic.edu; 5School of Public Health, University of Illinois Chicago, Chicago, IL 60612, USA; csando23@uic.edu (C.S.); amurph39@uic.edu (A.M.M.); 6Sinai Chicago, Chicago, IL 60612, USA; sarah.khan@sinai.org (S.K.); pam.khosla@sinai.org (P.K.); 7Equal Hope, Chicago, IL 60612, USA; paris_thomas@rush.edu; 8Sisters Working It Out, Chicago, IL 60612, USA; bbrent@sistersworkingitout.org (B.B.); lcoleman@sistersworkingitout.org (L.C.)

**Keywords:** health disparity, minority and vulnerable populations, cancer survivors, breast neoplasm, clinical trial, decision making, patient participation

## Abstract

Community-engaged research (CEnR) is a potent tool for addressing health inequities and fostering equitable relationships among communities, researchers, and institutions. CEnR involves collaboration throughout the research process, demonstrating improvements in study recruitment and retention, intervention efficacy, program sustainability, capacity building among partners, and enhanced cultural relevance. Despite the increasing demand for CEnR, institutional policies, particularly human participation protection training (HPP), lag behind, creating institutional barriers to community partnerships. Here, we highlight challenges encountered in our ongoing study, Fostering Opportunities in Research through Messaging and Education (FOR ME), focused on promoting shared decision-making around clinical trial participation among Black women diagnosed with breast cancer. Grounded in CEnR methods, FOR ME has a partnership with a community-based organization (CBO) that addresses the needs of Black women with breast cancer. Our CBO partner attempted to obtain HPP training, which was administratively burdensome and time-consuming. As CEnR becomes more prevalent, academic and research institutions, along with researchers, are faced with a call to action to become more responsive to community partner needs. Accordingly, we present a guide to HPP training for community partners, addressing institutional barriers to community partner participation in research. This guide outlines multiple HPP training pathways for community partners, aiming to minimize institutional barriers and enhance their engagement in research with academic partners.

## 1. Introduction

Community-engaged research (CEnR) is a potent tool for addressing health inequities and fostering equitable relationships between and among communities, community members, researchers, and research institutions. CEnR involves equitable collaboration throughout the research process with individuals from groups that the research will impact [1,2]. CEnR has significantly improved study recruitment and retention, efficacy, and effectiveness of interventions, program sustainability, capacity building among partners, and the enhanced cultural relevance of intervention and research questions [3,4].

Historically, a lack of trust has posed significant challenges for community partners, with research institutions often engaging in what is termed “drive-by” or “fly-in” research [5]. Considering this historical context, dedicating substantial time and effort to building and nurturing these relationships is critical. Financial disparities also present a major obstacle, leaving community partners feeling undercompensated for their vital contributions. Furthermore, the constraints of traditional research timelines can impede the development of meaningful relationships with key community figures, crucial for the success of CEnR. While the demand for CEnR approaches is rising, institutional policies supporting community members as research partners are lagging, creating barriers to community-academic partnerships. One such policy is the process of obtaining human participation protection training (HPP) for community partners who wish to engage in research processes.

Patients from racial and ethnic minorities are often diagnosed with cancer later and experience lower survival rates than non-Hispanic Whites due to multi-level social determinants of health [6,7]. The widening social, racial and ethnic health disparities observed over the past two decades highlight the urgent need for effective interventions and applied prevention research [8]. CEnR employs strategies to adapt interventions for diverse communities in cancer research, merging cultural knowledge with evidence-based socio-behavioral theory for comprehensive approaches. Increased attention to and implementation of community engagement in cancer research reflects a growing consensus on the importance of involving community members affected by research findings and of the availability of funding to support health equity [9]. For example, the National Institutes of Health’s (NIH) funding support for health disparities research has grown from $2.6 billion USD to $5.0 billion USD from 2008 to 2023 [10]. Community outreach and engagement (COE) has been recognized as a core activity of the National Cancer Institute’s (NCI) designated cancer centers since the inception of the program in 1971. In the 2016 and 2019 reissuances of the P30 Cancer Center Support Grant, COE became an explicit requirement for all NCI-Designated Cancer Centers [11]. All Centers are expected to develop, foster, and maintain bi-directional relationships with communities and implement CEnR across all programs—basic, clinical, translational, and population research [11,12]. While CEnR has been widely adopted as an evidence-based approach to improve healthcare outcomes, institutional policies that support integrating community members as partners in clinical research may pose challenges, especially when the level of collaboration calls for the community partner to complete institutional review board (IRB)-approved HPP training.

Our ongoing study, Fostering Opportunities in Research through Messaging and Education (FOR ME), aims to promote shared decision-making and, ultimately, participation in clinical trials among Black women diagnosed with breast cancer. The FOR ME study design is grounded in CEnR methods, and our team includes individuals from several organizations and institutions, including a community-based organization (CBO) focused on addressing the needs of Black women diagnosed with breast cancer, an academic medical research institution, a community safety net hospital, and a not-for-profit breast cancer advocacy organization. To move closer towards collective ownership on the spectrum of community engagement [13], we set collaboration goals with our CBO partner and determined that we would train a member of the CBO to lead a portion of the study’s qualitative data collection efforts. While the community partner was ultimately approved as key research personnel by our IRB, the process was administratively burdensome and time-consuming for our CBO partner. Upon reflecting on the challenges we experienced, we present a guide to human participant protection (HPP) training for community partners that promotes CEnR by addressing institutional barriers to community partner participation in research.

## 2. Challenges to Adopting Human Participant Protection Training for Community Partners

Recent experiences with the FOR ME project have highlighted the need to adapt the approval mechanisms of institutional regulatory bodies (i.e., IRB) to embrace CenR principles. Additionally, there is a need to offer HPP training to community partners that recognizes and addresses access and usability challenges associated with obtaining HPP certification. At a minimum, IRB HPP Collaborative Institutional Training Initiative (CITI) certification generally involves Human Subjects Research Training for biomedical, social, behavioral, and education researchers. These online, asynchronous, didactic trainings generally consist of required modules that span the historical context of the importance of HPP, federal regulations, components of informed consent, vulnerable populations, and internet-based research [14]. Case studies are included in each module to illustrate key concepts. Researchers who conduct clinical research are also required to complete Good Clinical Practice training, which consists of 9 online modules focused on applying sound clinical practice principles to clinical trial interventions [15]; and the Responsible Conduct of Research training, which consists of 12 online modules focused on authorship, collaborative research, conflicts of interest, human subjects, and research misconduct [16]. CITI cites the intended audiences for these trainings as human subject protection staff, IRBs, institutional/signatory officials, IRB administrators and staff, IRB chairs, researchers, and students [14]. Notably, community partners are not cited as intended audience members for these trainings. However, if we employ evidence-based CEnR approaches where community partners play an integral role in participant recruitment and interactions, they are required to take these trainings. Unfortunately, online HPP trainings are not adapted for diverse audiences who may significantly impact research processes yet are not traditional members of academia. For example, HPP trainings do not use the recommended literacy levels for public health communication and are not suitable for individuals without consistent digital access or comfort using computers and online platforms. Additionally, although the subject matters of HPP are critically important in protecting individuals participating in research, especially vulnerable populations, the time required to complete required online modules is extensive and may deter vital community partners from completing the trainings and, subsequently, participating in research processes.

In the FOR ME project, we recognized the importance of training community partners to participate in study recruitment, data instrument development and refinement, data collection, interpretation of results, and dissemination. Although our community partner was highly motivated and dedicated to serving as a research partner in this study, challenges related to the time commitment, communication about the appropriate training course, and administrative burden of completing the standard HPP training mechanism through online CITI training were encountered. Consequently, it took over three months from the time our community partner initiated administrative tasks (i.e., account setup and course assignment) for training to receipt of approval from our IRB. Fortunately, with the assistance of a dedicated regulatory staff member, our community partner navigated through this process, and it took about two months to complete the training courses and modules. This starkly contrasts with the CITI website, which states that the “average learner spends approximately 4.5 h to complete a Basic Course.” These multi-level systemic barriers contributed to project delays and created challenges with onboarding future community partners. Additionally, while CIRTification, an alternative community-based HPP training mechanism, is available through our institution, employing CIRTification is not a standardized process, and we faced challenges with using this mechanism due to questions about whether this training would provide appropriate HPP training for our community partner’s role in the project as key research personnel.

## 3. Building Effective Partnerships: Streamlining Regulatory Pathways for Community Engagement

As CeNR becomes more prevalent [9], academic and research institutions, as well as academic researchers, are faced with a call to action to become more responsive to community partner needs. Figure 1 offers a recommended decision tree for community partner HPP training and highlights the community partner, academic institution, and academic researcher factors that impact the ability of community partners to obtain HPP training.

During the initial phases of CEnR, academic researchers and community partners must assess the degree of community partner involvement, which ranges from research question development to dissemination. Through this process, an additional assessment should be made on whether community partners will interact with human participants, and thus would require HPP training. While traditional HPP training, such as online CITI training, is the standard HPP pathway at academic institutions, a CEnR-based approach would allow alternative training at recommended literacy levels for public health communication and in-person trainings for those without consistent broadband and digital access. Developed out of the University of Illinois Chicago (UIC), the CIRTification program is an alternative HPP training program that is tailored to the unique roles of community research partners and has been used nationally for CEnR [17,18]. CIRTification allows for in-person and online training in a manner that addresses the unique context of community-engaged research. While CIRTifcation offers a more CEnR-based approach to HPP training, the program must be recognized and accepted by academic institutions as an alternative training mechanism, and a consistent and standardized process for employing it must be established. A researcher must also check with their institutions if adapted HPP training programs, such as CIRTification, would be acceptable for each unique research project and community partner engagement. Finally, for those institutions where CIRTification or an equivalent is not recognized, allowing for a third pathway that uses academic researchers as community trainers would accommodate the unique roles that community partners play in CEnR. This customized training pathway would require upfront communication between the academic researcher and institution, approval of the training materials, and a commitment from the researcher to train community partners on HPP.

## 4. Conclusions

The journey of integrating CEnR into the fabric of clinical cancer research, as shown by the FOR ME project, underscores the imperative to reimagine and reform current institutional approaches to HPP training. The path forward necessitates not only acknowledging but actively accommodating the unique needs and contributions of community partners in the research process. By adopting more inclusive, flexible, and accessible training methods, such as the CIRTification program, and establishing alternative, CEnR-aligned pathways, academic institutions and academic researchers can adopt a “meet people where they are” approach to HPP training. This evolution in training and regulatory frameworks is essential to harnessing the true potential of CEnR, ensuring that research is not only conducted for the community but with the community, thereby enriching the relevance, impact, and ethical grounding of our scientific endeavors.

## Figures and Tables

**Figure 1 ijerph-21-00236-f001:**
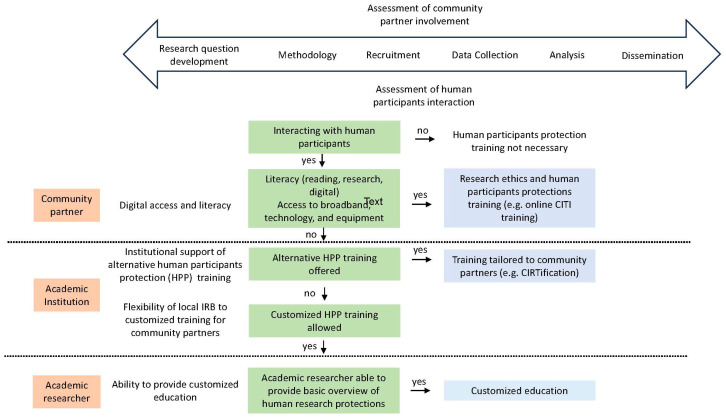
A decision aid for HPP training for community partners engaged in CEnR.

## Data Availability

No new data were created or analyzed in this study. Data sharing is not applicable to this article.

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
