# Peer review of "Are We the Problem? A Call to Action for Addressing Institutional Challenges to Engaging Community Partners in Research"

_ijerph, 2024, doi:10.3390/ijerph21020236_

Round 1

Reviewer 1 Report

Comments and Suggestions for Authors

Overall

Research through Messaging and Education (FOR 24 ME) project to incorporate community-engaged research (CEnR) practices. CEnR best practices for meaningfully engaging with community-based organizations (CBOs) includes engagement in human participation protection training (HPP). The commentary focused on the CBO partner’s attempted to obtain HPP training, which was administratively burdensome and time-consuming. The authors outline multiple pathways to overcome institutional barriers to obtain HPP training for community partners to ensure community partner participation in research.

Overall, the commentary is well written and organized and provides both interesting and relevant information to make equity real in research processes and as an evidence-based approach to improve healthcare outcomes.

Comments related to Figure 1 and Discussion Section 3. Building Effective Partnerships: Streamlining Regulatory Pathways for Community Engagement:

-For Figure 1, it would be helpful to understand how this process was developed. Is it solely based on experience of this research project? Did the research team use other publications to inform the development of this decision tree?

-For Figure 1, is “Community-Academic Based Research Partnership” the title for the figure or is this a heading about the research process? If title, I suggest to authors to make Figure 1 caption consistent with title of figure.

-In lines 110 and 111, the authors document how “HPP trainings do not use recommended literacy levels for public health communication”. However, this challenge is not included in the decision aid figure (currently only includes digital literacy). It would be helpful to understand any processes the research team implemented to overcome language accessibility issues with HPP training, if any.

Comments on the Quality of English Language

Overall, the commentary is well written and organized and provides both interesting and relevant information to make equity real in research processes and as an evidence-based approach to improve healthcare outcomes. A light round of copy editing is needed to fix a few typos.

Reviewer 2 Report

Comments and Suggestions for Authors

Dear authors, 

The skillfully crafted commentary deserves commendation for its adept language and insightful analysis. However, the absence of a definitive response to the crucial question posed under the headline, "Are we a problem or not?", creates a significant lacuna in the discourse. This lacuna compels further exploration in an academic tone, considering both potential answers and their corresponding implications. If the answer is "No, we are not a problem": Such a stance necessitates robust justification.

The commentary must: Clearly define "we" within the context of the discussion. Does it encompass humanity as a whole, specific demographics, or perhaps even non-human entities? The scope of the analysis hinges on this definition. Elaborate on the criteria for determining "a problem". What are the benchmarks by which success, sustainability, or harmony are measured?

Are these criteria objective or subjective, internal or external? Nuance in this regard is crucial. Present evidence-based arguments to support the assertion that "we" are not a problem. These arguments could highlight positive contributions to the environment, advancement of knowledge, or resolution of global challenges. Empirical data and sound reasoning would be vital to bolster this position.

If the answer is "Yes, we are a problem": Acknowledging this stance demands proactive solutions. The commentary should then: Identify the specific ways in which "we" pose a problem. Are we endangering ecosystems, exacerbating social inequalities, or jeopardizing our own future? Specificity fosters a targeted approach to solutions. Propose concrete and actionable solutions to address the identified problems. These solutions could encompass technological innovations, policy changes, or shifts in societal values. Feasibility and potential impact should be considered. Engage in a critical evaluation of potential drawbacks associated with the proposed solutions. Addressing alternative perspectives and potential unintended consequences demonstrates intellectual rigor.
